# SELF-SUPERVISED POLICY ADAPTATION

## ABSTRACT

We consider the problem of adapting an existing policy when the environment representation changes. Upon a change of the encoding of the observations the agent can no longer make use of its policy as it cannot correctly interpret the new observations. This paper proposes Greedy State Representation Learning (GSRL) to transfer the original policy by translating the environment representation back into its original encoding. To achieve this GSRL samples observations from both the environment and a dynamics model trained from prior experience. This generates pairs of state encodings, i.e., a new representation from the environment and a (biased) old representation from the forward model, that allow us to bootstrap a neural network model for state translation. Although early translations are unsatisfactory (as expected), the agent eventually learns a valid translation as it minimizes the error between expected and observed environment dynamics. Our experiments show the efficiency of our approach and that it translates the policy in considerably less steps than it would take to retrain the policy.

## 1 INTRODUCTION

Model-free reinforcement learning (RL) achieved remarkable results in the recent past and surpassed human performance on a number of complex tasks that have previously been considered intractable in applications such as playing games (Mnih et al., 2015), and created new opportunities in areas such as control in robotics (Schulman et al., 2015). Such agents directly interact with the environment and do not rely on a model for the environment as input.

While model-free (end-to-end) RL does not require *a priori* information of the (environment) model and its dynamics, it usually comes with a number of disadvantages. First, model-free RL suffers from *sample inefficiency*: it is not uncommon that it takes millions of training samples and trajectory rollouts to converge to a good policy. Second, as end-to-end RL trains a policy directly on raw observations it is virtually impossible to provide any *interpretation* of the policy and the agent's decision strategy (at least for high-dimensional input). Third, a trained policy that might already be running for a long time and that has been proven successful in a real-world application cannot simply be *transferred to work with a changed environment or different sensory input* (consider for instance a robot whose sensors break down or whose sensor outputs degrade or change, e.g., by radiation). Environment changes are also problematic for model-based agents. The first problem is well-studied and usually addressed with simulation-to-reality-transfer (i.e., train in a simulator and later transfer to the real world, possibly taking care of model mismatch) and the second one is a comparably novel research field where we might make use of methods such as model extraction (Bastani et al., 2017), saliency maps (Greydanus et al., 2018), or PIRL (Verma et al., 2018).

However, yet there is no common-sense approach to address changes to the environment, and in particular to the environment representation. Hence, in practice this often requires the policy to be retrained from scratch. If a system has already been deployed to the real-world application a retraining of the policy becomes even harder as we may not use elaborate techniques such as auxiliary tasks (Jaderberg et al., 2017), reward shaping (Ng et al., 1999), or hindsight experience replay (Andrychowicz et al., 2017), which could have been used in a laboratory setting to improve initial policy search. We cannot expect perfect extrinsic reward signals from the real world.

To the best of our knowledge the adaptation of an existing agent to changes in an environment representation (without leveraging the value function or the underlying reward process) still remains a vastly untouched research field and we set out to provide a first approach that shows promising results.

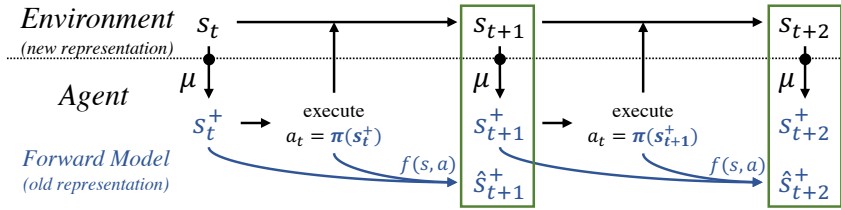

Figure 1: Greedy State Representation Learning.

Only recently Caselles-Dupré et al. (2018) took state representations from variational autoencoders (VAEs) and use generative replay (Shin et al., 2017) to align the latent state representations that has been used before an environment representation change to the one that is used after this change. However, the application of this approach is limited in practice as it requires a new autoencoder to be trained that captures all generative factors of the environment upon the change. In practice, we need to iteratively train and acquire new samples while partially exploring the environment to obtain comprehensive information about the generative factors (approaches that use VAEs always need to bootstrap (Ha and Schmidhuber, 2018)) unless the observation itself describes the entire environment, e.g. if the representation is a top-view of a maze that needs to be navigated.

The most simplistic idea is to acquire labeled samples from the environment (states for which we know both the old and the new encoding), i.e., state-pairs, and use them to train a model that translates the state representations. However, we cannot obtain such pairs of state representations as (i) the old raw input is no longer available, (ii) potentially stored observations from early runs (old encoding) cannot be matched to the new observations, and (iii) we cannot expect that an *oracle* or a human provides manual translations for observations as the representation is usually too complex.

*Greedy State Representation Learning* (GSRL) obtains such state-pairs indirectly and uses them to train a neural network model that serves as a translator for observations obtained after the environment change into the representation corresponding to the environment before the change, so that we can re-use policies trained for the initial environment representation. Figure 1 sketches the general idea. The environment (top) works on the new representation while the agent (below the dashed line) works on the old representation (blue). Using an initially randomly initialized model $\mu$ we translate a new state observation $s_t$ from the environment into its old representation $s_t^+$ (noisy translation), which we use to select an action $a_t$ from the earlier learned policy $\pi$. We now use the obtained action twice. First, we query a forward model $f$ (that captures the environment dynamics and that we have trained with the data acquired before the environment representation changed) to get an estimate of the expected next state $\hat{s}_{t+1}^+$ in the old representation, i.e., our biased state estimation *target*. Second, we execute the action $a_t$ in the environment and obtain the new state representation $s_{t+1}$ of the next state. We translate this new representation with the neural network model $\mu$ into $s_{t+1}^+$ and calculate a loss that we use to minimize the error relative to our expected state estimation target $\hat{s}_{t+1}^+$ such that the translations come close to the target estimated by the forward model. Our experiments on a `mountaincar` environment show the efficiency of our approach and that we can bootstrap a model that translates state representations in less iterations than it would take to retrain the policy.

The remainder of this paper is organized as follows. Section 2 discusses related work. Section 3 recalls basic definitions and foundations from reinforcement learning. Section 4 formalizes the considered problem and introduces our GSRL algorithm. Section 5 describes the experimental setup and Section 6 discusses the results.

## 2  RELATED WORK

*Classic approaches* that are commonly used in sensor fusion such as Kalman or particle filters only provide limited applicability for extreme changes to the representation. Instead, *fault-tolerant control* that uses, for instance, Gaussian processes and model-based controllers (e.g., model predictive control, MPC) (Yang and Maciejowski, 2015), provides more flexibility. However, such approaches also struggle with severe changes to the representation and can no longer reliably control such situations. Cully et al. (2015) propose a two-step algorithm to adapt after severe damage (e.g. a robot losing a limb), that (1) *a priori* calculates a behavior map that captures the expected performance of behavioral

models, and (2) evaluates those behaviors upon damage. However, as this requires to enumerate and calculate the maps of all possible (partial) breakdowns *a priori* (which is usually intractable) it does not scale both in memory and computation time to real world applications. Approaches such as AdaPT (Zero-Shot Adaptive Policy Transfer for Stochastic Dynamical Systems) (Harrison et al., 2017) compensate a dynamics mismatch with MPC to attenuate the bounded dynamics between source and target dynamics. However, AdaPT does not assume that the representation changes.

*Domain randomization* assumes small discrepancies between a source and a target domain. Added perturbations in training ensure that the policy does not overfit to the training environment and will later generalize well in the target domain. These approaches adapt the policy in the target domain (Daftry et al., 2016), use adversarial training (Pinto et al., 2017; Mandlekar et al., 2017) with perturbed dynamics, assume and compensate for dynamics mismatches (Muratore et al., 2018), or adapt the reward that is provided by the environment (Wang et al., 2018; Romoff et al., 2018). However, they only work for mild changes to the environment but do not cope with severe changes (such as completely missing elements).

In contrast, *domain adaptation* trains on a particular input distribution with a specified reward structure (source domain) and then adapts the agent to a modified input distribution with the same reward structure (target domain). DARLA (Higgins et al., 2017b) learns disentangled generative factors of the environment (with $\beta$-VAEs (Kingma and Welling, 2014; Higgins et al., 2017a)) and uses this latent variable representation to learn a policy in the source domain. However, DARLA is no solution if perturbations severely affect the encoding of the generative factors (which is our assumption). Tzeng et al. (2017) address domain shift and adaptation through *weak* pairs of visual input from two domains with similar structure and elements. A combined minimization of task, confusion, and pairwise loss makes the policy robust to domain shifts. The similarity to our approach lies in the acquisition of labeled training input from unlabeled observations but this approach assumes a similarity between the images, which we cannot expect in our setting.

There is also a connection of the presented work here to recent work in continual learning and multi-task learning (Parisotto et al., 2016). E2C (Watter et al., 2015) derives a latent state space representation based on VAEs in which dynamics are locally linear to apply locally robust optimal control. While E2C *translates* the state representation to a latent space for linear control it does not address changes in the state representation. Finn et al. (2017) use semi-supervised RL to train an agent on a set of tasks in environments where a reward function is available and use inverse RL to generalize to unknown environments. However, while they deal with changed dynamics and rewards (as for the different tasks) they are not concerned with large, systematic domain shifts and changes to the environment representation. Gupta et al. (2017) propose a multi-agent transfer learning in a setting where two (physically different) agents learn multiple skills. For skills that have been learned by both agents, each of them constructs a mapping from their observed states to an invariant feature space. With a richer feature space an agent can learn a new skill by projecting the executions of the other agent into its own feature space. While a common feature space would also be a solution for the problem at hand this approach cannot be applied as (upon a sudden change to the representation) there is no longer a contribution to the feature space by the agent that works on the old representations.

The work closest related to ours is Caselles-Dupré et al. (2018) who build a state representation model with VAEs. Upon detection of a change to the environment (but not to the underlying dynamics) they sample from the latest VAE using generative replay (Shin et al., 2017) and then train an updated model together with new samples acquired from the environment. However, there are challenges depending on the environment and the exploration as the agent needs to iteratively use a greedy policy and sample all relevant areas of the environment (Ha and Schmidhuber, 2018). If the agent does not see the whole environment at once this approach is eventually equivalent to retraining the policy.

## 3 PRELIMINARIES

We consider the standard reinforcement learning formalism consisting of an agent interacting with an environment. To simplify the notation and without loss of generality we assume that the environment is fully observable, i.e., that $o_t$ is a fully observed realization of the (true and fully described) environment state $s_t$ at time step $t$. A *Markov decision process (MDP)* is described by a set of *states* $\mathcal{S} \in \mathbb{R}^n$, a set of *actions* $\mathcal{A} \in \mathbb{R}$, a distribution of initial states $p(s_0)$, a *reward function* $r : S \times A \to \mathbb{R}$, *transition probabilities* $p(s_{t+1}|s_t, a_t)$, and a *discount factor* $\gamma \in [0, 1]$.

A *deterministic policy* is a mapping from states to actions: $\pi : \mathcal{S} \to \mathcal{A}$. Every episode starts with sampling an initial state $s_0$. At every time step $t$ the agent produces an action based on the current state: $a_t = \pi(s_t)$. Then it receives the reward $r_t = r(s_t, a_t)$ and the environment's new state is sampled from the distribution $p(\cdot|s_t, a_t)$. A discounted sum of (future) rewards is called a *return*: $R_t = \sum_{i=t}^{\infty} \gamma^{i-t} r_i$. The agent's goal is to maximize its expected return $\mathbb{E}_{s_0}[R_0|s_0]$. The *Q-function* or action-value function is defined as $Q^\pi(s_t, a_t) = \mathbb{E}[R_t|s_t, a_t]$.

Let $\pi^*$ denote an *optimal policy*, i.e., any policy $\pi^*$ s.t. $Q^{\pi^*}(s, a) \geq Q^\pi(s, a)$ for every $s \in \mathcal{S}, a \in \mathcal{A}$ and any policy $\pi$. All optimal policies have the same $Q$-function which is called *optimal Q-function* and denoted $Q^*$. It is easy to show that it satisfies the *Bellman equation*:

$$Q^*(s, a) = \mathbb{E}_{s' \sim p(\cdot|s,a)} \left[ r(s, a) + \gamma \max_{a' \in \mathcal{A}} Q^*(s', a') \right].$$

In practice, the $Q$-function will be approximated with value function approximation (Mnih et al., 2015), which also allows to scale RL to high-dimensional state spaces such as raw sensory input from cameras in an end-to-end setting. As an alternative to an end-to-end approach we can also make use of a function $q_\phi(s_t)$ that, given a set of parameters $\phi$, returns a compact representation of features that are relevant for an agent to control its effect on the environment. For instance, VAEs (Kingma and Welling, 2014) have been demonstrated to learn structured latent representations of high dimensional data and also have been applied to RL (Higgins et al., 2017b; Ha and Schmidhuber, 2018).

## 4 GREEDY STATE REPRESENTATION LEARNING

### 4.1 PROBLEM FORMULATION

Now let $\pi : o_t \to a_t$ be a policy that has been trained using some model-free RL algorithm, see Figure 2. In fact, the observation $o_t$ is generated by an underlying function $o_t = g(s_t)$ that returns sensor information that we assume that fully describes the hidden environment state $s_t$ (if $o_t$ is not Markovian we may use methods such as frame stacking (Mnih et al., 2015) or recurrent networks to ensure this). In other words, $g$ turns a state into a *state representation*. In practice, such functions $g$ are e.g., implemented in cameras to compute pixel arrays from light input. If $o_t$ is large or complex, e.g., a camera image, we may also apply some post-processing on it (e.g., given by a VAE's encoder that extracts the latent variables from the observations) or some manually engineered feature descriptors that provide a more concise

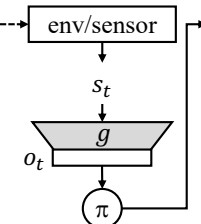

Figure 2: MDP setting.

representation of $o_t$. Those transformations can be applied directly on $o_t$ but we omit them to simplify the notation. We can use $o_t = g(s_t)$ to sample an action from a given policy $\pi$. In the following our assumption is that $\pi$ is a non-trivial complex, robust, and well-proven (or even *certified*) control policy for which a re-estimation is considered costly. As such, upon a change to the environment, our goal is to reuse the policy $\pi$ rather than to retrain it from scratch.

Now let us assume further that (e.g., due to some unexpected event) the hidden process/function $g(s_t)$ that generates $o_t$ changed and we instead obtain some perturbed version $g^+(s_t) = o_t^+$. In a real-world sensor system this might be caused by (a) damages of single sensors (e.g., dead pixels in camera frames) or by (b) loss of functionality of a sensor in a multi-sensor system (if all the sensor input is collected and provided to the agent in an end-to-end RL setting). We still assume that $o_t^+$ is a Markovian realization of the hidden state $s_t$ such that the MDP assumption still holds (in (a) the sensor output is perturbed but still unique and in (b) other sensors might compensate for the damage). However, under the assumption that $o_t$ and $o_t^+$ are too different we cannot expect to use the policy $\pi$ to sample actions (and probably we neither can apply the same preprocessing as we did it on $o_t$).

Our goal is to infer a *translation model* $\mu$ with parameters $\psi$ that converts the observation $o_t^+$ to its original value $o_t$, both for the same hidden state $s_t$ so that we can reuse the policy. In other words, $\mu_\psi$ should translate between the distributions that generate the observations from the states. An easy solution is to let the agent sample the environment by following its policy and then to use pairs of observations $(o_t, o_t^+)|s_t$ to train a model using supervised learning. However, as the agent only sees

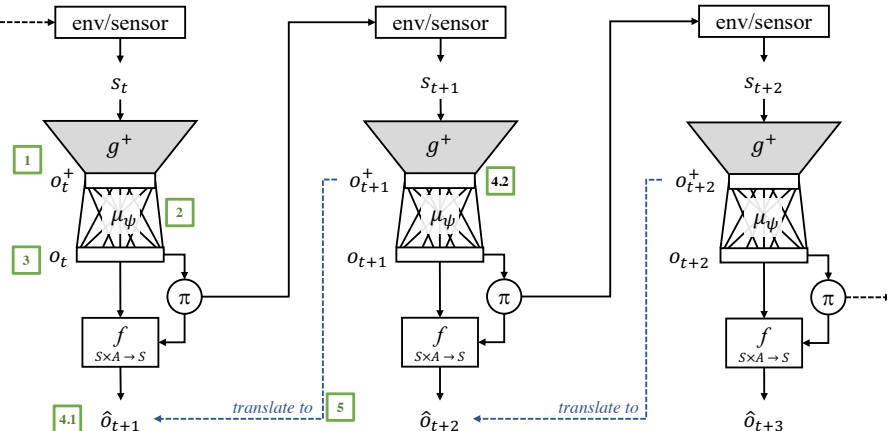

Figure 3: Translating the state representation: (1) the new observation is provided by the environment (and optionally further preprocessed); (2) a neural network model translates the observation to its old pendant; (3) using this observation we sample an action $a_t$ from policy $\pi$; (4.1) through the forward model $f$ we obtain a prediction for the succeeding observation $\hat{o}_{t+1}$ and (4.2) by sampling the environment we obtain the succeeding observation $o_{t+1}^+$; (5) we train $\mu_\psi$ with $\left[ o_{t+1}^+; \hat{o}_{t+1} \right]$.

observations from a single distribution at a time (either from before *or* after the incident) it will never receive both observations for a single $s_t$ rendering this easy solution impractical.

## 4.2 MATCHING BIASED PREDICTIONS TO REAL OBSERVATIONS

We want to find parameters $\psi$ that minimize the translation error according to a distance metric $d$ (i.e., *mean squared error*, *huber-loss*, *l2-loss*, etc.) over samples of observations from the environment $E$ that we obtain by executing actions $a_t$ (given the environment's state $s_t$):

$$\psi = \underset{\psi \in \mathbb{R}^d}{\arg\min} \; \mathbb{E}_{o_t^+ \sim E(\cdot, a_t), a_t \sim \pi} \left[ d\big( o_t, \mu_\psi(o_t^+) \big) \right].$$

This is a relaxed version of the original problem of translation as it only translates observations from the environment that are actually relevant to the agent. This is why it is sufficient to calculate the expectation over samples that we acquire by following policy $\pi$. Note that in particular more frequently sampled actions implicitly receive a high weight in the objective.

The key idea is to estimate a neural network model $\mu$ with a set of weight parameter $\psi$ that we can use for state translation: $\mu_\psi(g^+(s_t)) \approx g(s_t)$ and that converts the new state representation to the old state representation. While initial policy training (or original operation of the agent) we use observed transitions $[g(s_t), a_t, g(s_{t+1})]$ to train a neural network forward model $f$ with parameters $\theta$ that captures the dynamics of the environment:

$$g(s_{t+1}) = f_\theta\big( g\left(s_t\right), a_t \big), \tag{1}$$

in order to predict the next observation. We use this forward model to predict *biased targets* of succeeding observations $\hat{o}_{t+1}$ that we then can use together with the altered environment observation $o_{t+1}^+$ (i.e., those observations obtained after the state encoding has changed) to train our translation model. Forward models have been proven to work well in practice but may suffer for larger planning horizons, i.e., if they are used to predict several consecutive states.

We illustrate the setup in Figure 3. We observe a new observation $o_t^+ = g^+(s_t)$ that we might have further preprocessed using an updated encoder. We translate $o_t^+$ into $o_t$ using an initially randomly initialized model $\mu_\psi$. This gives us the observation from the initial distribution over $s_t$, i.e., a noisy estimate of $g(s_t)$. Next, we use $o_t$ to select an (optimal w.r.t. the translated state/observation) action $a_t$ from our given policy $\pi$.

Now, we make use of $a_t$ along two routes. First, we use the forward model $f_\theta(o_t, a_t)$ to predict a biased estimate of the next observation $\hat{o}_{t+1}$ (according to the old distribution $g$). This provides the

state that the agent expects to end up in when it follows the policy under the old state representation. Second, we actually execute this action to obtain the real next observation (according to the current distribution $g^+$) from the environment: $o_{t+1}^+ = g^+(s_t), s_t \sim E_{a_t \sim \pi}$. This way we obtain a pair $(o_{t+1}^+, \hat{o}_{t+1})$ that we can push to a replay buffer where we later sample mini-batches to train the translation model.

The intuition behind this is that for two succeeding states the translations along the routes result in the same original state representation, i.e.,

$$\underbrace{f_\theta\left[\mu_\psi\big(g^+(s_t)\big), a_t\right]}_{\text{via forward model (left)}} = \underbrace{\mu_\psi\left[g^+\big(E(s_t, a_t)\big)\right]}_{\text{via environment (right)}}. \tag{2}$$

To further simplify we first multiply the observation provided by the environment $E$ on the right side with the identity, i.e., $g$ and its inverse $g^{-1}$:

$$f_\theta\left[\mu_\psi\big(g^+(s_t)\big), a_t\right] = \mu_\psi\left[g^+\Big(g^{-1}\big(g\big(E(s_t, a_t)\big)\big)\Big)\right]. \tag{3}$$

We assume that $f_\theta$ ideally captures the environment dynamics, i.e., $f_\theta(o_t, a_t) = g\left[E(s_t, a_t)\right]$. Hence, instead of sampling the observation from the environment $E$ we sample it from the forward model $f_\theta$:

$$f_\theta\left[\mu_\psi\big(g^+(s_t)\big), a_t\right] = \mu_\psi\left[g^+\Big(g^{-1}\big(f_\theta(o_t, a_t)\big)\Big)\right]. \tag{4}$$

Next, we can further reformulate the right side and eliminate $\mu_\psi$ as $\mu_\psi(o_t^+) = o_t = g\big((g^+)^{-1}(o_t^+)\big)$ (i.e., the inverse of $g^+$ applied on $o_t^+$ gives $s_t$ and $g$ applied on $s_t$ gives $o_t$):

$$f_\theta\left[\mu_\psi\big(g^+(s_t)\big), a_t\right] = g\left[(g^+)^{-1}\Big(g^+\big(g^{-1}\big(f_\theta(o_t, a_t)\big)\big)\Big)\right] \tag{5}$$

$$\Rightarrow f_\theta\left[\mu_\psi\big(g^+(s_t)\big), a_t\right] = g\left[g^{-1}\big(f_\theta(o_t, a_t)\big)\right] \tag{6}$$

$$\Rightarrow f_\theta\left[\mu_\psi\big(g^+(s_t)\big), a_t\right] = f_\theta(o_t, a_t). \tag{7}$$

We can easily reformulate the left side, as $g^+(s_t) = o_t^+$ and $\mu_\psi(o_t^+) = o_t$:

$$f_\theta(o_t, a_t) = f_\theta(o_t, a_t). \tag{8}$$

Note that we do not assume the invertibility of $g^+$ and $g$ in practice as we never actually invert them. However, for the relaxed problem space both $g^+$ and $g$ are at least locally invertible as this is ensured by the Markov property of the state representation.

### 4.3 Matching Ambiguous Dynamics Manifolds

Training the parameters $\psi$ for translator $\mu$ with labeled pairs $(o_t^+, o_t)$ is problematic when applied in practice. While the objective of the training problem captures the desired goal there might be still (local) ambiguity in the dynamics over the statespace that can pose significant challenges. To exemplify this, consider a one-dimensional state-space with states $x \in \mathbb{R}$ and the underlying dynamics from Figure 4 (at state $x$ the observation is

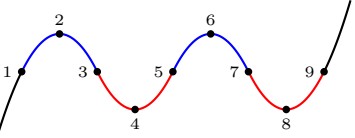

Figure 4: Ambiguous Dynamics.

$g(x) = x$ and the dynamics $f$ behave according to the given function). Assume that $g^+(x)$ is now different from $g(x)$ and we instead receive some perturbed version of $g(x)$. Our algorithm tries to find a mapping such that relative changes between observations behave as expected, i.e., *we try to find a mapping such that the expected dynamics (given by the forward model) match the actually observed dynamics.*

In Figure 4 the dynamics in $[1; 3]$ are equal to those in $[5; 7]$ (similar for $[3; 5]$ and $[7; 9]$). A (surjective) mapping of pairs of observations $(o_t^+, o_{t+1}^+)$ from both regions $[1; 3]$ and $[5; 7]$ to either one of those in the original encoding will result in zero (target) translation error (only the transitions of the incorrectly mapped trajectory at the (left) borders of this regions will have a non-zero translation error) as the translations suffer from a *local view* on the dynamics.

We address the ambiguity of dynamics with two counter-measures. First, we exploit the Markov property of the observations: instead of only optimizing $\mu_\psi$ for the translation error we add a

reconstruction loss regularizer that enforces that a target representation, i.e., the old encoding, can be translated back to the given representation, i.e., to the new encoding. This adds a penalty if different $o_t^+$'s are mapped to the same $o_t$. For this we use a decoder $\mu^{-1}$ with parameters $\nu$ that takes the output $o_t$ of $\mu_\psi$ and decodes it to $\tilde{o}_t^+ \approx o_t^+$. We jointly train the parameters $\psi$ and $\mu$ using stochastic gradient descent and standard backpropagation using the loss function:

$$\mathcal{L}(o_t^+, \hat{o}_t; \psi, \nu) = L_\delta \left[ \hat{o}_t - \mu_\psi \left( o_t^+ \right) \right] + \beta \cdot L_\delta \left[ o_t^+ - \mu_\nu^{-1} \left( \mu_\psi \left( o_t^+ \right) \right) \right], \qquad (9)$$

where $L_\delta$ is the huber loss function, the first term is the target translation error, the second term is the reconstruction error, and $\beta$ is a hyperparameter that balances the influence of the reconstruction error over the translation error.

Our second counter-measure is a lookahead extension that makes use of multi-step transitions. Instead of generating translation targets from single steps through the forward model and the environment we perform $\lambda > 1$ steps into the future. This gives $\hat{o}_{t+1}, \ldots, \hat{o}_{t+\lambda}$ (note that the actions $a_t, \ldots, a_{t+\lambda}$ are not sampled from the policy using $\hat{o}_i$ but based on the immediate translations $o_i$). We can reinitiate this process of generating translation targets starting from $o_{t+1}$ with $\lambda$ then reaching out to $o_{t+\lambda+1}$. We add the additional samples to the replay buffer as we also did single-step predictions. See Appendix A for more details.

In essence, the interaction between the forward model and the environment by following our policy provides labeled data pairs that we can use to train $\mu_\psi$. If the system dynamics are constant everywhere, i.e., for all states and any actions the derivative of $f$ is constant (in each dimension), then there are infinitely many solutions for parameters $\psi$ for which Equation 2 holds. For instance, consider a simple cartpole, where the state is described by the position of the cart (among other variables). Aside from the environment which decides if an episode ends the actual dynamics of the system are independent of the position. If we assume an infinite rail we would not be able to translate the actual position of the cart. The same holds for any environment where dynamics are invariant to particular variables of the state representation. We cannot resolve this ambiguity as the translation error of the generated targets vanishes even for an arbitrary biased translation of this variable.

## 5 EXPERIMENTAL SETUP

**Environments and Baseline Policy.** To evaluate our method we used OpenAI's `MountainCar-v0` environment with discrete actions $a \in [\texttt{left}; \texttt{noop}; \texttt{right}]$ and a two-dimensional state space that is defined by the position $\in [-1.2; 0.6]$ and the velocity $\in [-0.07; 0.07]$ of the car. We train a dueling Double-DQN baseline policy (Hessel et al., 2018) with a single hidden layer with 64 units and ReLU activations and a replay buffer size holding 50,000 transitions (using prioritized experience replay with $\alpha = 0.6$, $\beta_0 = 0.4$). We use a learning rate of $1e-3$ with the ADAM optimizer, a batch size of 32, and apply parameter noise. We update the target network every 500 time steps and let the agent train for 500k time steps.

As we experienced high variance among different runs for both the baseline policy training and our state translation we changed the exploration behavior to better focus on the algorithmic behavior. Figure 5a shows the mean episode reward over the last 100 episodes of different runs of our vanilla DQN agent using an $\epsilon$-greedy exploration schedule ($\epsilon$ drops linearly from 1 to 0.01 in 10k time steps ($\approx 50$ episodes)). Depending on the randomness we get different performance for the runs.

Instead of using an $\epsilon$-greedy schedule to foster exploration we use a static $\epsilon = 0.02$ and extend the range for the initialization of the environments, i.e., such that the states may take any value from within the state space. Hence, for the monitoring and evaluation of the training process we use two different environments: (i) a training environment (with custom initialization), and (ii) an evaluation environment (with original initialization and a greedy agent where we can clearly measure the achieved reward). Figure 5b shows the episode reward for this custom policy evaluation. Despite of the collapse of the reward, which is a well-known issue and expected for value function approximation (van Hasselt et al., 2018)), we see a much more robust performance across the runs. We will use this custom environment initialization for all our experiments.

**Forward Model Architecture.** To train the forward model we use the observations that the agent collects during initial policy training of our DQN baseline. We concatenate the input $o_t$ and $a_t$ into a single vector (as we use the version with discrete actions we used a one-hot encoding) and pass it into an multilayer perceptron (MLP) with 3 layers having 256 units each. We use ReLU activation

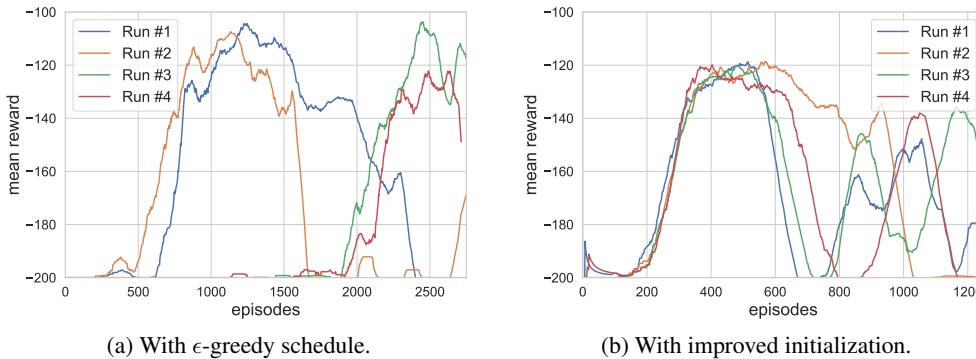

(a) With $\epsilon$-greedy schedule.

(b) With improved initialization.

Figure 5: DQN baselines on OpenAI gym's `MountainCar-v0`.

for the hidden units and linear activation to provide the output. To improve the robustness of the forward model we only predict updates, i.e., given $o_t$ and $a_t$ as input the forward model predicts the output $o_{t+1} - o_t$, which reduces the prediction error for unseen samples and hence results in a better generalization (Deisenroth and Rasmussen, 2011). As the dimensionality of the inputs and outputs is defined by the environment we normalized and shifted both the input and output accordingly. We do not apply any regularization or dropout. The loss is calculated using a mean squared error, the network is optimized using the ADAM optimizer and a learning rate of $1e - 3$. We trained 150 epochs with a batch size of 32.

**Translation Model Architecture**. The translation uses two sub-models: a translation and a reconstruction model. The former maps the input observation $o_t^+$ into $o_t$ and the latter reconstructs $o_t^+$ from $o_t$. In our experiments both use an MLP with a single hidden layers having 128 units. We apply *tanh* activations to the outputs of the hidden layers and a linear activation to the output layers (i.e., for the translation and for the reconstruction). We do not use any regularization or dropout. We initialize the weights with Xavier and the biases with zeros. For the loss function we use the huber loss with $\delta = 1.0$ and apply $\beta = 1.0$ for the reconstruction error regularizer. We use a batch size of 1.

We also use a replay buffer with size 10,000 where we sample 16 mini-batches at each time step. We use target networks (i.e., copies of $\mu$ and $\mu^{-1}$) that we update after any 500 time steps. For lookaheads $\lambda > 1$ we generate all the targets for $\lambda = 1, ..., n$ and push them to the replay memory.

The source code will be made available for download upon publication.

## 6 RESULTS

**Translating the State Representation.** We perturb the state representation (we tried many and they result in similar performance): instead of $[\texttt{position}, \texttt{velocity}]$ the environment provides the simple distortion $[\texttt{velocity} \cdot 2, \texttt{position}/2]$. We run an agent that takes a DQN baseline policy (one from Figures 5b) and that interacts with the environment to obtain data that we process according to Section 4. Figure 6a shows the mean (and standard deviation) of the episode rewards for both 10 runs of our algorithm with $\lambda = 10$ and the DQN baseline from Figure 5b) according to our policy evaluation scheme. We reach convergence after approx. 300 episodes. For any of those runs our algorithm translates the state representation such that the agent can reuse the given policy. To achieve this, we need considerably less samples that we would need to retrain the policy from scratch. At the same time our method never sees any ground truth of both the state representation or the reward.

Note that there are also small reward drops for our translator over some episodes. We noticed that (for the evaluation environment) even small errors in the translation (of the velocity) around the starting point of the cart lead to a different action sampled from the policy: the agent gets stuck in the valley next to the base of the hill as it mistakenly thinks that it has some velocity that allows it to go further up the hill. The episode then terminates at a reward of $-200$, i.e., after 200 time steps. Indeed, we see very good and accurate translations for most of the runs even after 100 episodes in training.

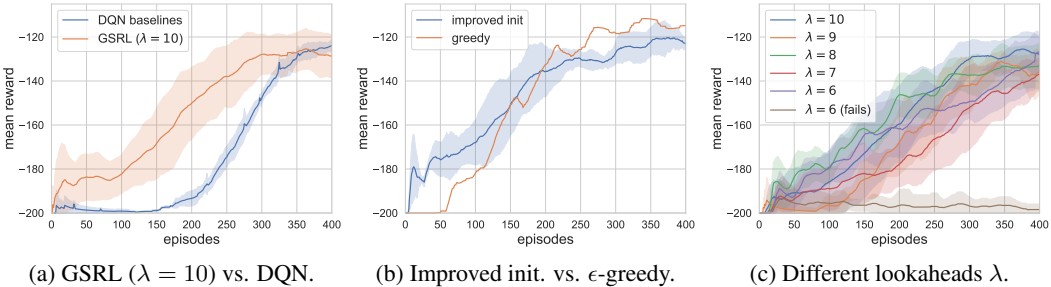

(a) GSRL ($\lambda = 10$) vs. DQN.     (b) Improved init. vs. $\epsilon$-greedy.     (c) Different lookaheads $\lambda$.

Figure 6: GSRL on the `MountainCar-v0` environment.

For the sake of completeness Figure 6b shows a run with an $\epsilon$-greedy schedule (orange) and a standard evaluation setup of the environment (only one environment, $\epsilon$-greedy policy) next to the custom evaluation (blue). While we see the early rise later the $\epsilon$-greedy policy reaches the plateau *en par* with the improved initialized environment. However, the performance varies between the different runs due to the randomness that is introduced by the exploration-exploitation schedule.

**Lookahead Study.** We also compared the influence of the lookahead parameter $\lambda$ on the performance of the translation. To capture the influence more clearly we did not use the forward model here as we want to leave out any effects from the accuracy of the forward model for longer prediction horizons (as for larger prediction horizons error accumulate). Instead, we here use an implementation of the actual dynamics of the `MountainCar-v0` environment and whenever we query the forward model we instead initialize a fresh environment with the observation, step through the environment using $a_t$ provided by $\pi$, and take the resulting state/observation as a prediction of the forward model. Figure 6c shows the mean reward and standard deviation over episodes for different values of $\lambda$ (10 runs for each). In general we see that larger prediction horizons result in faster/better convergence and (although hardly visible) the variance between the runs decreases. Interestingly, the high variance targets that are generated especially from early samples (resp. from their noisy representations) do not have a negative impact on the performance. For $\lambda \leq 6$ we observed that for 50% of the runs the translation gets stuck in a local optimum where it cannot escape (this is due to the effects described in Section 4.3). The other 50% of the runs behave similar to those for larger $\lambda$'s.

In practice we need to adjust $\lambda$ carefully as there is a trade-off between convergence, accuracy of the forward model, and the dynamics manifold. Hence, $\lambda$ should be chosen based on a hyperparameter optimization. Based on this it is also conceivable to use a schedule that starts with a low $\lambda$ (as we have high variance in the translations in the beginning) that is steadily increasing as the translations of the targets get less noisy (this would then allow to also escape local minima).

**Policy Improvement.** Figure 7 shows the policy on the translated input over the training of our translator at the beginning, after 5k, 10k, 25k, and 40k iterations/steps. The $x$-axis denotes the (true) position within $[-1.2; 0.6]$ and the $y$-axis denotes the (true) velocity within $[-0.07; 0, 07]$. We generate the images as follows. We initialize the internal state of the environment according to each point and retrieve its perturbed version. We translate this perturbed observation using the current translator and sample the policy for an action. We draw the actions at the respective coordinates and encode them with blue circles (`left`), green rectangles (`noop`), and orange crosses (`right`). The most right image shows an almost perfect translation of the perturbed observations with respect to the underlying policy, i.e., to gain momentum and if velocity is positive the agent applies `right`, and if it is negative it applies `left`. Interestingly, the translations lead to the correct action for the majority of the state-space even after 10k iterations (approx. 50-60 episodes). For the remaining iterations the agents only *fine-tunes* the translations. Hence, depending on the actual problem/environment at hand the agent might reach a sufficient translation very early. The rightmost translation of the policy is almost identical to the original policy (this is why we omit the original policy).

We refrain from showing graphs that plot the error of the translated samples (both against ground truth and targets) as they behave as expected: the error of the model against the ground truth is initially large and decreases towards 0 while the error against the targets starts from a lower value and decreases more slowly (as targets are biased).

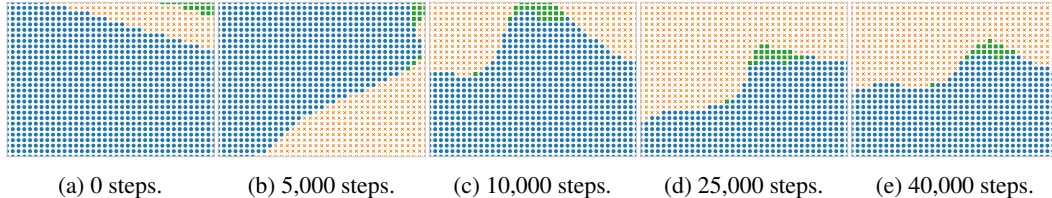

(a) 0 steps.   (b) 5,000 steps.   (c) 10,000 steps.   (d) 25,000 steps.   (e) 40,000 steps.

Figure 7: *Translated policy* emerging over iterations. Encoding: `left` (blue circles), `noop` (green rectangles), `right` (orange crosses). The rightmost policy is almost identical to the original policy.

## 7  CONCLUSION

This paper proposed a novel algorithm that translates an environment representation using a given policy. To the best of our knowledge this has been the first attempt to address this problem. We sample observations from both the environment and a pretrained forward model and bootstrap a translator that minimizes the differences between expected and actually observed system dynamics. While we used an RL-based policy our method is agnostic to the controller, e.g., it also works with optimal control or MPC. Most importantly, our approach can also be used when there is no reward process and no ground truth samples available. We would also like to point out that, while we did not explicitly test for this our method also works for state representations that are generated by VAEs and hence can be applied to high-dimensional input (however, training the VAE on a new environment representation requires careful exploration and bootstrapping). During our work we also came up with a much more simplistic solution to the given problem, see Appendix B. However, it turned out that this approach has significant downsides compared to the one we presented in this paper.

In future work we intend to investigate the influence of stochasticity of the environment to the policy transfer process. While this is very well studied in terms of the effects on a forward model (Racanière et al., 2017) stochasticity affects our approach twice (in the training of the forward model and in the generation of targets). Another interesting direction is to apply a curiosity-based exploration scheme (Pathak et al., 2017) to generate more informative translation targets.

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

## A  LOOKAHEAD EXTENSION

Figure 8 sketches the idea behind multi-step predictions for our translation. The elements above the dashed line show a simplified version of Figure 3: we translate any observation $o_t^+$ into $o_t$; the policy $\pi$ gives us $a_t$; we obtain $o_{t+1}^+$ from the environment and $\hat{o}_{t+1}$ from the forward model; and we use $(o_{t+1}^+; \hat{o}_{t+1})$ to train the parameters $\psi$ for translator $\mu$.

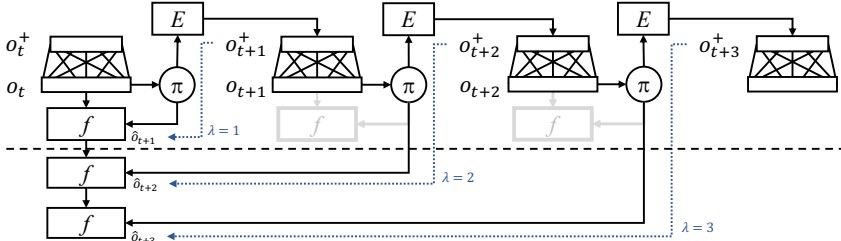

Figure 8: Creating translation targets through multi-step predictions.

Similar to eligibility traces ($n$-step prediction) known from TD-learning (Sutton and Barto, 1998) we can generate targets by looking $\lambda$ steps into the future. For instance, in the lower part in Figure 8 we take $\lambda = 3$ steps through the environment and the forward model (the original idea uses $\lambda = 1$, i.e., one step through the forward model), which gives us $\hat{o}_{t+1}$, $\hat{o}_{t+2}$, $\hat{o}_{t+3}$ (note that the actions $a_t$, $a_{t+1}$, $a_{t+2}$ are not sampled from the policy using $\hat{o}_i$ but based on the immediate translations $o_i$). We can reinitiate this process of generating translation targets starting from $o_{t+1}$ with $\lambda = 3$ then reaching out to $o_{t+4}$. We add the additional samples to the replay buffer as we also did single-step predictions.

## B  ALTERNATIVE APPROACH TO TRANSLATE STATE REPRESENTATIONS

A much simpler alternative that is very similar and that we also investigated is depicted in Fig. 9. The key idea of this approach is an inverse dynamics model, i.e, a model that gives the action that has been taken to make a transition from $o_t$ to $o_{t+1}$. Such models have been investigated to learn features for recognition tasks (Pathak et al., 2017; Agrawal et al., 2015; Jayaraman and Grauman, 2017). We train this model on transitions $(o_t, a_t, o_{t+1})$, i.e., we use $\mu$ to translate $o_t^+$ to $o_t$ and sample our policy $\pi$ to obtain $a_t$, which we execute in the environment. We receive $o_{t+1}^+$ and translate it to $o_{t+1}$. We use the *inverse* forward model $f^{-1}$ on $(o_t, o_{t+1})$ to predict $\hat{a}_t$. We use gradient descent to minimize the distance between $a_t$ and $\hat{a}_t$ and regularize the reconstruction errors on $o_t$ and $o_{t+t}$ similar to Section 4.

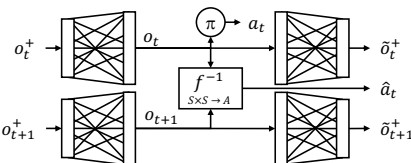

Figure 9: Alternative Transcoding Scheme.

However, in practice this approach does not work well because of numerous reasons. The noisy estimates we get out of $\mu$ are likely to represent invalid state transitions. The inverse model cannot generalize to such unseen (and impossible) state transitions and returns a highly biased action $\hat{a}_t$. Even if the estimations from $\mu$ would become more accurate the inverse model will eventually introduce a bias. As the inverse model cannot work on state updates (i.e., predicting only the delta between states) but instead must be trained to predict a full state representation it will be not as accurate. This leads to a problem when it comes to generalization error for previously unseen states. Hence $a_t$ will always have some non-negligible bias that will render this method unstable. In contrast to the approach from Section 4 we cannot use lookaheads to trade variance for bias.

