# OpenReview forum: "Self-Supervised Policy Adaptation"
_ICLR.cc/2020/Conference — Reject_

### Official Review · AnonReviewer3 · 2019-10-15
**Official Blind Review #3**

**Rating:** 1

**Review:**

The paper investigates a setting in which the observation function changes while the underlying environmental dynamic stays the same.
In order to re-use the policy which was trained on the old observation function, they propose to learn a mapping function to map the new observations to the old ones.

I believe the work is interesting as generalization and reducing the sample complexity of learning policies, for example through re-use of old policies, is of high current interest.  However, I believe this paper requires more work to show the feasibility of the proposed approach. In particular:
- The proposed method has the problem that matching is done 'locally' and without any guarantee that the mapping function will converge to the correct mapping. This is not a problem of the method itself but of the challenging problem setup. The authors discuss this and propose two approaches to alleviate this. However, no experimental evidence is shown whether and to what extend this prevents wrong local minima, as I don't believe the mountain car has such problems?
- Evaluation is only done on the mountain-car experiment, which is not sufficient to show feasibility in general
- The learning curves in the experiment seem highly volatile and unstable. I'm not sure why this is the case, maybe wrong hyperparameters or a bug in the code?
- Lastly, the paper is over the recommended limit of 8 pages and could, in my opinion, made more concise at many points to shorten it and also make it easier to read.

In summary: I believe this interesting work, but requires more experiments in different environments and additional ablation studies to show the feasibility of the proposed method. Making the writing more concise would, in my opinion, not only shorten the paper but also make it easier to read.

**Experience Assessment:**

I have published one or two papers in this area.

**Review Assessment: Checking Correctness Of Derivations And Theory:**

I assessed the sensibility of the derivations and theory.

**Review Assessment: Checking Correctness Of Experiments:**

I assessed the sensibility of the experiments.

**Review Assessment: Thoroughness In Paper Reading:**

I read the paper at least twice and used my best judgement in assessing the paper.

---

### Official Review · AnonReviewer2 · 2019-10-23
**Official Blind Review #2**

**Rating:** 1

**Review:**

The authors propose a means to adapt to new state representations during reinforcement learning. The method works by learning a translation model that translates new state representations to old state representations. The authors evaluate the method on the MountainCar environment and show that the adaptation model is more efficient than training a new policy from scratch.

My concerns with this work are as follows:

- I don't find the type of changes to state representations very useful (e.g. Section 6, velocity *=2, position /= 2) nor practical. Moreover, learning a translation model to recover the old representation seems easy given this type of simple perturbations. This perturbation is fundamentally different than those used to motivate the problem (e.g. "a robot whose sensors break down" or "whose sensor outputs degrade", in the introduction).
- The authors only evaluate with one type of simple perturbation on one environment, hence I am skeptical regarding the generalizability of this work.
- The premise of this work is to not store old transitions (Section 1, the paragraph "the most simplistic idea is to..."), however this model does store old transitions because it uses prioritized experience replay. In this case there is no argument against using this data to train the translation model.
- A comparison that is missing from the paper is to fine-tune the existing model. I believe this is a more fair comparison in terms of sample-efficiency than training from scratch.


Other comments:
- For the title, to call the adaptation to slightly different state representations "policy adaptation" is a stretch.
- There are a lot of tangential information in the introduction on things like sample efficiency and model-free vs. model-based RL. This is distracting.
- The paper is excessively long.

**Experience Assessment:**

I have published one or two papers in this area.

**Review Assessment: Checking Correctness Of Derivations And Theory:**

I assessed the sensibility of the derivations and theory.

**Review Assessment: Checking Correctness Of Experiments:**

I assessed the sensibility of the experiments.

**Review Assessment: Thoroughness In Paper Reading:**

I read the paper at least twice and used my best judgement in assessing the paper.

---

### Official Review · AnonReviewer1 · 2019-10-26
**Official Blind Review #1**

**Rating:** 1

**Review:**

NOTE: This paper is 10-pages long which requires a higher bar according to the guidelines.

Summarize what the paper claims to do/contribute.
* This paper claims to propose the first method to translate an environment representation to a different representation when that changes.

Clearly state your decision (accept or reject) with one or two key reasons for this choice.
Reject.
* The results were not adequate. I suggest exploring more environments and more complex ones than MountainCar.
* I do not believe this is the first attempt in translating an environment represention to a different one. Other techniques in domain adaptation have been working on this for quite some time. It might be the case that the technique proposed here is better eg to pixel-based adaptation or adaptation of features, but it would need to be shown experimentally.
* Given the higher number of pages, i would have expected more thorough experimentation.

**Experience Assessment:**

I have published in this field for several years.

**Review Assessment: Checking Correctness Of Derivations And Theory:**

I assessed the sensibility of the derivations and theory.

**Review Assessment: Checking Correctness Of Experiments:**

I carefully checked the experiments.

**Review Assessment: Thoroughness In Paper Reading:**

I made a quick assessment of this paper.

---

### Author Response · Authors · 2019-11-15
**Rebuttal Comment**

We would like to thank the Reviewers for their feedback. Unfortunately, we were not able to perform additional experiments and address all comments to the extent and depth we would like to within the rebuttal period, but we will use the feedback as a guideline for improving on our paper and results and resubmit in the future.

---

### Decision · Program_Chairs · 2019-12-19

**Decision:**

Reject

**Comment:**

The submission proposes to improve generalization in RL environments, by addressing the scenario where the observations change even though the underlying environment dynamics do not change. The authors address this by learning an adaptation function which maps back to the original representation. The approach is empirically evaluated on the Mountain Car domain.

The reviewers were unanimously unimpressed with the experiments, the baselines, and the results. While they agree that the problem is well-motivated, they requested additional evidence that the method works as described and that a simpler approach such as fine-tuning would not be sufficient.

The recommendation is to reject the paper at this time.